# Uricase-Expressing Engineered Macrophages Alleviate Murine Hyperuricemia

**DOI:** 10.3390/biomedicines12112602

**Published:** 2024-11-14

**Authors:** Yu-Zhong Feng, Hao Cheng, Guo-Qing Xiong, Jia-Zhen Cui, Zhi-Li Chen, Yuan-Yuan Lu, Zhi-Xin Meng, Chen Zhu, Hao-Long Dong, Xiang-Hua Xiong, Gang Liu, Qing-Yang Wang, Hui-Peng Chen

**Affiliations:** 1Academy of Military Medical Sciences, Beijing 100071, China; torty@163.com (Y.-Z.F.); 18323215576@163.com (H.C.); 18225910118@163.com (G.-Q.X.); cjiazhen2023@163.com (J.-Z.C.); chenzl0708@163.com (Z.-L.C.); zhuchen521521@163.com (C.Z.); haolong_dong@163.com (H.-L.D.); xxianghuabj@163.com (X.-H.X.); 15701121020@163.com (G.L.); 2Institutes of Physical Science and Information Technology, Anhui University, Hefei 230000, China; luyuanyuan1205@163.com; 3School of Life Science, Hebei University, Baoding 071000, China; mengzhixin68@163.com

**Keywords:** uricase, hyperuricemia, macrophage therapy, engineered macrophages

## Abstract

**Background**: Uricase, or urate oxidase (Uox) is a key enzyme in uric acid (UA) metabolism and has been applied in clinical treatment of human hyperuricemia (HUA). However, the current clinically applied uricases, despite their potent urate-lowering capacity, tend to form anti-drug antibodies because of their immunogenicity, leading to increased risk of anaphylaxis, faster drug clearance and reduced or even complete loss of therapeutic effect, limiting their clinical application. In this study, we constructed engineered macrophages that stably expressed uricase, which might serve as a promising alternative to the direct injection of uricases. **Materials and Methods**: Engineered macrophages RAW264.7 cells were injected intravenously to treat hyperuricemic KM mice. Serum uric acid and bio-indicators for renal and hepatic functions were detected by an automatic biochemical analyzer; inflammatory cytokines were determined by ELISA; the livers and kidneys of the mice were sectioned for histological examination. **Results**: The uricase-expressing macrophages reduced UA levels from 300 ± 1.5 μmol/L to 101 ± 8.3 μmol/L in vitro. And in an HUA mouse model established by gavage with yeast extract, intravenous injection of the engineered macrophages could reduce the serum uric acid (sUA) of mice to normal level on the 14th day of modeling, with a decrease of 48.6%, and the urate-lowering effect was comparable to that of the first-line clinical drug allopurinol. In terms of safety, engineered macrophages did not cause liver or kidney dysfunction in mice, nor did they induce systemic immune response. **Conclusions**: Using macrophages as a chassis to deliver uricase might be a new, safe and effective strategy for the treatment and control of hyperuricemia.

## 1. Introduction

Hyperuricemia (HUA) is a metabolic disease that is caused by impaired purine metabolism and reduced excretion of uric acid (UA), characterized by elevated serum uric acid (sUA). Long-term excessive sUA tends to lead to the formation of monosodium urate (MSU) crystals in articular and periarticular regions and results in gout [1,2]. In addition, clinical studies demonstrate that HUA is also a risk factor for cardiovascular and cerebrovascular diseases, diabetes, hypertension, hyperlipidemia, metabolic syndrome and chronic renal insufficiency [3,4,5].

Conventional therapies for the treatment of HUA are mainly divided into three categories. The first category is uricosurics, represented by probenecid and benzbromarone, which promote the excretion of UA by inhibiting the re-absorption of urate in the renal tubules. The second category is xanthine oxidase inhibitors, represented by allopurinol and febuxostat, which hinder the purine metabolic pathway by inhibiting the activity of xanthine oxidase to reduce the production of UA [6,7]. The third category is recombinant uricases, which catalyze uric acid into allantoin that is 5–10 times more soluble than UA and is more easily excreted by the kidneys [8]. The only two marketed recombinant uricases are rasburicase and pegloticase. Uricases have more potent urate-lowering effects and less adverse reactions; thus, they have become a research hotspot in the treatment of hyperuricemia.

Rasburicase is a recombinant uricase that originates from *Aspergillus flavus* and is expressed in *Saccharomyces cerevisiae*, which was approved by the FDA in 2002 [9], and pegloticase is a PEG-modified recombinant porcine–baboon chimeric uricase which was approved by the FDA in 2010 [10]. Uricases can quickly and significantly reduce the level of sUA, but the currently marketed uricases are heterologous, which have high immunogenicity, leading to the formation of anti-uricase antibodies and therefore causing anaphylaxis, infusion reactions and treatment failure [11,12]. For example, rasburicase has a homology of less than 40% with the deduced human uricase, and antibodies are easily produced after re-administration, so this treatment is only approved for short-term treatment of tumor lysis syndrome (TLS)-associated acute hyperuricemia [11]. Pegloticase is a long-acting biological protein drug modified by polyethylene glycol (PEG). Compared with rasburicase, it has a longer half-life and lower immunogenicity and can be used for long-term treatment of refractory gout [13]. Although the immunogenicity of pegloticase is significantly lower, as use of the medication continues, antibodies against PEG are still produced in up to 89% of patients, resulting in a gradual decrease in the urate-lowering effect, and 58% of patients experience a loss of effectiveness after multiple injections [14]. Therefore, pegloticase is often co-administered with immunomodulators such as methotrexate to achieve the expected therapeutic effect [15], which also limits the long-term use of this drug. Structural optimization and immunomodulator co-administration both have their limits and cannot diminish the immunogenicity of exogenous proteins. We speculate that reducing direct immune exposure of uricase may become a promising strategy for alleviating immunogenicity.

Various drug delivery systems have been researched in recent years. SEL-212, which has now entered phase III clinical trials, is a yeast-derived pegylated recombinant uricase (Pegadricase) encapsuled with an immunotolerant-inducible drug delivery platform (nanoparticles containing rapamycin, ImmTOR). It can inhibit the formation of ADA and reduce the immunogenicity of the drug [16,17]. He et al. constructed an *Escherichia coli* Nissle 1917 (EcN) that expressed a functional uricase. Through oral administration, it significantly reduced the blood uric acid level in hyperuricemic rats [18]. Various types of vectors are being developed to deliver uricase, including nanoparticles [16,17,19], probiotics [18], erythrocyte [20], macrophage membrane [21,22], etc., which all have unique advantages.

Cell therapy is an emerging treatment approach for conditions such as hematological [23], autoimmunological [24] and neurological diseases [25]. It involves extracting autologous or allogeneic cells, performing genetic modifications on them in vitro, and then re-injecting them into patients to exert their therapeutic effects. With great successes achieved by T cell therapy and NK cell therapy, cell therapy utilizing macrophages has also attracted increasing interest from researchers [26]. Macrophages, characterized by properties like inflammatory tropism and a long half-life, have innate advantages as a tool for cell therapy. Macrophages can be engineered to express exogenous genes and then be injected back into the body for therapeutic purposes. To date, manipulation approaches to macrophage engineering have become increasingly mature and diverse, with the available approaches including electroporation transfection [27,28], viral-mediated transfection and cytokine-mediated reprogramming [29,30], which have improved the operability of macrophages at the genetic level. In addition, the improvement of in vitro expansion [31,32] and pluripotent stem cell (PSC) directional differentiation technologies [33,34] has greatly enhanced the feasibility of macrophage therapy. Macrophage-based cell therapy has been proven to be an effective method of tumor control in animal models. Ray et al. used CRISPR-Cas9 gene editing technology to knock out the signal regulatory protein-α (SIRP-α) gene in macrophages, leading to a four-fold increase in phagocytic ability against tumor cells [35]. Morrissey et al. modified macrophages through chimeric antigen receptors to enhance their phagocytic ability against tumor cells [36]. Li et al. enhanced the targeting and phagocytic ability of macrophages through loaded nanoparticles and surface modification technologies [37]. FU et al. constructed a biomimetic delivery system (BDS) by loading doxorubicin into RAW264.7 macrophages, which showed promising anti-cancer efficacy and low toxicity for lung metastasis of breast cancer in a mouse model [38]. These studies show a broad application prospect of macrophage-based cell therapy. However, macrophage therapy has not been applied to the treatment of hyperuricemia.

In this study, we used mouse-derived macrophage RAW264.7 cells as the chassis and used lentiviral transfection to integrate the mouse uricase gene (*mus*-Uri) into the cell genome to construct engineered macrophages. The expression profile of uricase shows that wild type RAW264.7 expresses very low levels of uricase, which makes it an ideal chassis for exogenous expression of uricase. Since uricase is encapsulated in macrophages, they can avoid direct encounter with the immune system, thus reducing its immunogenicity. Furthermore, we verified the in vivo urate-lowering activity of engineered macrophages on hyperuricemic mice established by gavage with yeast extract and evaluated its safety for the liver and the kidney and its effects on the immune response.

## 2. Materials and Methods

### 2.1. Strains, Plasmids and Vector Construction

The *mus*-Uri gene sequence was codon-optimized and synthesized by Tsingke Biotechnology (Beijing, China), and constructed into the lentiviral vector pLVX-puro by enzyme digestion and ligation. pLVX-puro carries a green fluorescent protein expression cassette, which can be used for screening and enriching positive cells. The plasmid was transfected into *Escherichia coli* DH5α, and positive clones were selected for sequencing. The correctly edited plasmid was commissioned to Ubigene Biosciences to construct the lentiviral vector. The lentiviral vector carrying the *mus*-Uri gene was labeled as LV-*mus*-Uri; the control lentivirus was labeled as LV-Ctrl.

### 2.2. Cells and Lentiviral Transfection

Mouse macrophage RAW264.7 was obtained from the Cell Resource Center, Peking Union Medical College (PCRC), and cultured with DMEM medium supplemented with 10% fetal bovine serum (Gibco) and 1% penicillin/streptomycin solution (Gibco) and maintained at 37 °C with 5% CO_2_. RAW264.7 cells in logarithmic growth phase were seeded in a 6-well culture plate at a density of 7 × 10^5^ cells/well, and LV-*mus*-Uri lentivirus and LV-Ctrl lentivirus were added to the cells according to the optimal MOI (Multiplicity of Infection) recommended by Ubigene Biosciences, and polybrene was added at a final concentration of 5 μg/mL. The cells continued to be cultured in the incubator. After 24 h of viral transfection, the culture medium containing the virus and polybrene was removed and replaced with fresh DMEM complete culture. After 48 h of viral transfection, the cells were observed via fluorescence microscopy, and the transfection efficiency was detected and calculated by flow cytometry. The macrophages after lentiviral transfection were labeled as RAW-*mus*-Uri and RAW-Ctrl, respectively.

### 2.3. qPCR

The expression of the uricase gene in the RAW-*mus*-Uri cell samples was detected by qPCR, using RAW-Ctrl as the negative control and mouse liver cells as the positive control. Total RNA was extracted using the TRIzol kit from Ambion (Austin, TX, USA). Probes and primers were designed according to the sequences of *mus*-Uri and the reference gene GAPDH. The expression difference in the target gene in each group of cells was calculated according to the Ct value.

### 2.4. Western Blot

Engineered macrophages RAW-Ctrl and RAW-*mus*-Uri were lysed and the protein concentration was determined by the BCA method. After undergoing denaturing at 100 °C for 5 min, protein samples were subjected to SDS-PAGE gel electrophoresis and then transferred onto the NC membrane. After blocking with milk for 1 h, the membranes were incubated with the primary antibody overnight at 4 °C, followed by incubation with the secondary antibody at room temperature for 1 h. The enhanced chemiluminescence (ECL) reagent was evenly dripped onto the membrane and the bands were visualized by a chemiluminescence imaging system.

### 2.5. In Vitro Evaluation of Urate-Lowering Activity

Urate-lowering activity of engineered macrophages in DMEM culture medium was evaluated as follows: RAW-Ctrl and RAW-*mus*-Uri were seeded in a 12-well culture plate and incubated in an CO_2_ incubator for 6 h to stabilize the cell state, and then UA solution was added into the wells at a final concentration of 300 μmol/L. After that, 100 μL supernatant samples were taken every 12 h for 72 h, and the UA concentration was measured with an automatic biochemical analyzer.

### 2.6. Engineered Macrophage Treatment of Hyperuricemic (HUA) Mice

A total of 50 KM male mice (18–22 g) were purchased from Beijing Vital River Laboratory Animal Technology (Beijing, China) and raised in the animal breeding room of the Academy of Military Medical Sciences under SPF conditions. The mice were randomly divided into 5 groups (n = 10). One group of mice was used as the control group (Ctrl) and did not undergo any intervention. The other 4 groups were gavaged with 20 g/kg/d yeast extract for 14 days to establish the HUA mouse model. Among the 4 groups of mice given yeast extract, the HUA group was given yeast extract only, the allopurinol group was gavaged with allopurinol at a dose of 20 mg/kg/d for 14 consecutive days, and the other 2 groups were injected with RAW-Ctrl and RAW-*mus*-Uri via the tail vein on the 1st and 7th days, respectively, receiving a dose of 100 μL each time, at a concentration of 2 × 10^7^ cells/mL. Tail clip blood samples were obtained from all mice on days 0, 7, and 14, and the sera were collected after centrifugation, and the serum UA level of the mice was detected by a biochemical analyzer.

### 2.7. Hepatic/Renal Function and Inflammatory Factors Evaluation

The levels of alanine aminotransferase (ALT), aspartate aminotransferase (AST), total bilirubin (TBIL), direct bilirubin (DBIL), creatinine (CREA), and urea (UREA) in the serum of the mice in the abovementioned groups were detected by a biochemical analyzer. The inflammatory factors IL-6, IL-10 and IL-1β in the serum were detected by ELISA with commercial kits purchased from Dakewe Biothch Co., Ltd (Shenzhen, China).

### 2.8. Histopathology

The mice in the abovementioned groups were sacrificed by cervical dislocation at day 14, and the kidneys were obtained and fixed in 4% paraformaldehyde for 24 h. The tissues were fixed, paraffin-embedded, sectioned, stained with hematoxylin and eosin (H&E), and incubated at 37 °C for 24 h. The sections were scanned and imaged using Pannoramic DESK (3DHISTECH Ltd., Budapest, Hungary).

### 2.9. Statistical Analysis

GraphPad Prism 8.0 software was used for statistical analysis in this study. Student’s *t*-test was used to compare the differences between two groups of data; *p* < 0.05 was considered statistically significant.

## 3. Results

### 3.1. Engineered Macrophages Express Exogenous mus-Uri

The EGFP positive rate of lentivirus transfected macrophages was detected by flow cytometry. The results showed that the transfection efficiency was 48.4% for LV-Ctrl group, and 34.5% for the LV-*mus*-Uri group, indicating that the lentivirus successfully infected the macrophages (Figure 1A). Furthermore, the mRNA level of *mus*-Uri expressed by the two engineered macrophages was detected by qPCR, and the liver tissue, which has the highest expression level of Uri in mice, was used as the positive control. The results showed that RAW264.7-Ctrl cells did not express Uri, and RAW-*mus*-Uri significantly expressed *mus*-Uri, at a level comparable to that in the liver tissue (Figure 1B). Finally, a specific band was detected in the RAW-*mus*-Uri cells by Western Blot (Figure 1C), indicating that the exogenous uricase could be correctly expressed in the engineered macrophages.

### 3.2. Engineered Macrophages Lower UA Level In Vitro

The in vitro urate-lowering experiment showed that during the growth process in a uric acid-containing medium, the *mus*-Uri expressing macrophages RAW-*mus*-Uri reduced the UA level in the medium in a time-dependent manner. Within 72 h, the UA concentration decreased from 300 ± 1.5 μmol/L to 101 ± 8.3 μmol/L, with a reduction rate of 66.3% (Figure 2). This indicates that RAW264.7, when used as a chassis to express exogenous uricase, possesses in vitro urate-lowering activity and can be used for further in vivo evaluation studies. Additionally, the RAW264.7-Ctrl group did not show a urate-lowering capacity, ruling out the possibility that the RAW264.7 chassis or the lentivirus itself has urate-lowering effects.

### 3.3. Engineered Macrophages Lower Serum UA Level in HUA Mice

Before the in vivo experiment, RAW-*mus*-Uri cells were isolated and enriched for EGFP-positive cells through flow cytometric sorting. The results showed that the proportion of EGFP-positive cells was about 30% (Figure 3A), and after a single round of expansion culture, the number of cells obtained was sufficient for the animal experiment. The grouping and treatment of mice were as described in Section 2 (Section 2.6). The UA levels in each group showed no significant difference on day 0. On days 7 and 14 after gavage with yeast extract, the sUA levels in the HUA group mice increased from 120.6 ± 18.5 μmol/L to 193.2 ± 6.4 μmol/L and 210.0 ± 24.4 μmol/L, respectively, representing increases of 60.2% and 74.1% from the initial average, indicating that the continuous gavage with yeast extract successfully induced HUA in mice. The sUA levels in the RAW-Ctrl group increased by 44.5% and 49.5% on days 7 and 14, respectively, while the changes in the RAW-*mus*-Uri group were −5.3% and −11.0%, and the changes in the allopurinol group were −4.8% and −12.1% (Figure 3B, Appendix A). The results indicate that the RAW264.7 chassis itself did not exhibit urate-lowering activity, whereas compared to the HUA + RAW-Ctrl group, the HUA + RAW-Uri group had significantly lower UA levels on day 7 and day 14, with a urate-lowering effect comparable to that of the first-line clinical medication allopurinol.

### 3.4. Safety Evaluation of Engineered Macrophages

Using allopurinol as a control medicine, the safety of the engineered macrophages RAW-*mus*-Uri for the liver and kidney was evaluated. The liver function indicators revealed no significant differences between the treatment groups and the control, indicating that the engineered macrophages and allopurinol had no adverse effects on liver function. As for the kidney function indicators, crea and urea, no differences were observed for the RAW-*mus*-Uri group showed, whereas the allopurinol group showed 37.6% and 40.8% increases in these markers, respectively, compared to the control group (Figure 4A). No significant abnormalities were detected in the inflammatory factors IL-6, IL-10, and IL-1β in macrophage-treated groups (Appendix A). Histopathological evaluation revealed no obvious lesions in the livers of mice in all groups (Figure 4B); however, compared to the blank control group, the kidney tissues of mice in the allopurinol group displayed evident abnormalities, such as immune cell infiltration, renal cell necrosis, and hemorrhaging, whereas no obvious lesions were observed in the RAW-*mus*-Uri group (Figure 4C). These results suggest that the engineered macrophages do not have significant adverse effects on the liver and kidney functions of mice during treatment, exhibiting a better safety profile than allopurinol.

## 4. Discussion

Traditional treatments for hyperuricemia mainly include xanthine oxidase inhibitors and uricosurics. These medications have issues such as poor targeting, numerous side effects, low compliance, and long treatment durations. For example, febuxostat has been associated with an increased risk of cardiovascular-related death, leading to a black box warning for this medication being issued by the FDA [39]; allopurinol poses a risk of fatal allergic reactions in patients with risk factors, especially those carrying the HLA-B*5801 allele [40]; and benzbromarone is limited for wide application due to its rare but potentially lethal hepatotoxicity [41]. Therefore, it is crucial to seek new urate-lowering strategies and provide more treatment options.

Recombinant uricase, as an emerging treatment for hyperuricemia, offers a more potent ability to lower UA levels and is an ideal alternative for patients who are intolerant to traditional medications, have contraindications, or for whom treatment has been ineffective. However, uricase medications, as heterologous proteins, have certain limitations in terms of their clinical application. The incidence of allergic reactions to rasburicase after multiple doses reaches 6.2% [11], and although pegloticase has a lower incidence of allergic reactions, high titers of anti-PEG antibodies can develop after multiple infusions, leading to reduced efficacy or even complete loss of effectiveness [14].

To reduce the immunogenicity of uricase medications, past strategies mainly focused on modifying and optimizing its protein structure, such as designing new PEG chains or reducing the number of PEG chains on the protein molecule [42]. In spite of these efforts, no new drugs of this nature have been approved for clinical use. In this study, we used lentiviral transfection to introduce the mouse uricase gene *mus*-Uri into murine macrophage RAW264.7, creating an engineered macrophage that can express uricase in vivo. This method avoids direct exposure of the drug’s protein molecules to the host’s immune system, theoretically resolving the issue of immunogenicity. Moreover, macrophages are characterized by an ability to phagocytose urate crystals, a lifespan that can be as long as several months, and tropism towards inflammation, making them an ideal chassis for treating hyperuricemia. Both in vitro and in vivo experiments have validated that the engineered macrophages carrying the exogenous uricase gene can lower uric acid levels, with the in vivo urate-lowering effect after two administrations comparable to that of the first-line clinical medication allopurinol; assessments of liver and kidney functions and the measurement of inflammatory factors indicate that the engineered macrophages have a good safety profile in vivo. It should be noted that our research is limited, as we only used male mice for the research. The results in female mice might be complicated by the existence of estradiol [43].

The strategy of engineering macrophages in this study is still a long way from clinical application and requires further extensive research, including research that determines how to achieve engineered editing using autologous macrophages to minimize the immunogenicity and potential tumorigenic risks of exogenous cells, as well as a more comprehensive safety assessment of this therapy, including blood cell analysis, blood clotting function, and long-term safety studies. However, this study has explored the feasibility of using cell therapy to treat hyperuricemia. In the context of the limitations of existing treatment methods, this study provides a new therapeutic option.

## Figures and Tables

**Figure 1 biomedicines-12-02602-f001:**
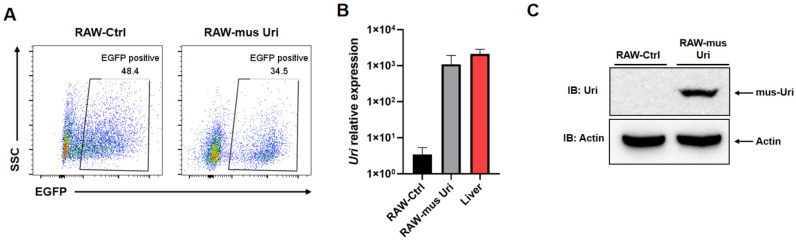
*mus*-Uri expression in the engineered macrophage RAW264.7. (**A**) Positive rate after lentiviral transfection, detected by flow cytometry; (**B**) uricase relative expression determined by qPCR; (**C**) *mus*-Uri protein expression confirmed by Western blotting.

**Figure 2 biomedicines-12-02602-f002:**
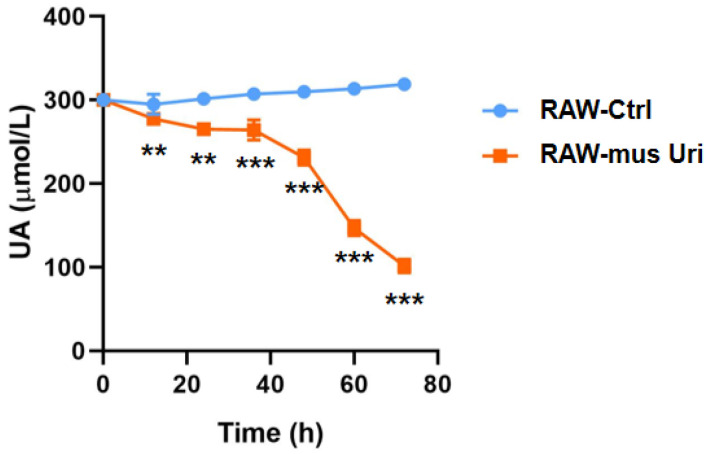
Engineered macrophage reduces UA level in vitro. Comparison was performed between RAW-mus Uri and RAW-Ctrl groups at each time point. (n = 10, ** *p* < 0.01, *** *p* < 0.001).

**Figure 3 biomedicines-12-02602-f003:**
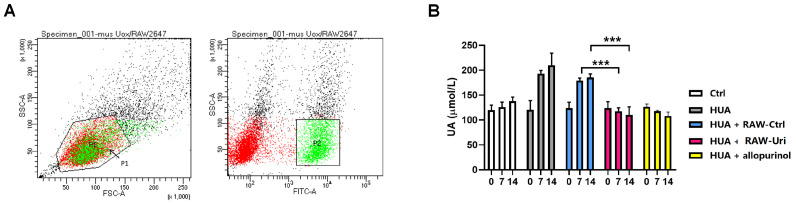
(**A**) Rate of EGFP-positive RAW264.7 after cytometry sorting; P1 represents all viable cells, while P2 represents EGFP-positive cells. (**B**) Uric acid levels in each group 0, 7, and 14 days after intervention. The levels between HUA + RAW-Ctrl and HUA + RAW-Uri were compared. (n = 10, *** *p* < 0.001).

**Figure 4 biomedicines-12-02602-f004:**
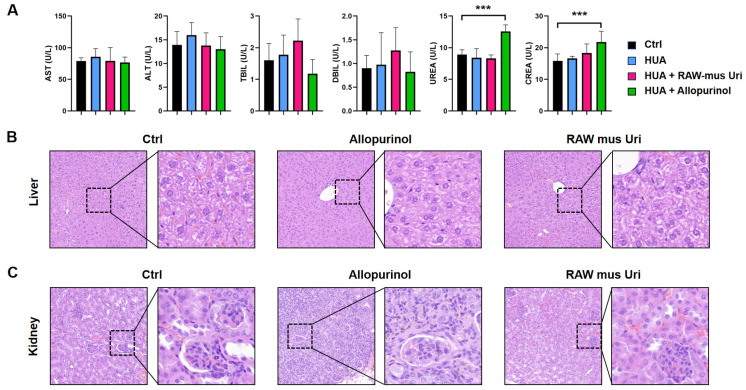
Safety evaluation of engineered macrophage RAW-*mus*-Uri. (**A**) Hepatic function indicators AST, ALT, TBIL and DBIL and renal function indicators urea and crea collected from serum are assessed. (n = 10, *** *p* < 0.001) (**B**,**C**) H&E staining and histological examination of liver (**B**) and kidney (**C**) in each group.

## Data Availability

The original contributions presented in the study are included in the article/Appendix A. Further inquiries can be directed to the corresponding author.

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
