# Peer review of "Uricase-Expressing Engineered Macrophages Alleviate Murine Hyperuricemia"

_biomedicines, 2024, doi:10.3390/biomedicines12112602_

Round 1

Reviewer 1 Report

Comments and Suggestions for Authors

Reviewer comments: 

1.      Why authors use the NC membrane instead of PVDF membranes for the western blot.

2.      Student's t-test was used to compare the differences between the two groups of data. It is suitable for the data of Figure 2. However, data of Figure 3B has more than 2 groups. Therefore, ANOVA and post-hoc should be applied to calculate the statistical differences between groups.

3.      In line 221-224, authors indicated that macrophages expressing the exogenous mouse uricase gene mus-Uri reduced the UA levels below the initial levels. However, Figure 3B shows only statistical difference between the UA levels on day 7 and day 14 but not day 0. I also do not find the comparison between HUA + RAW-Uri group and Ctrl group. Authors should show these statistical differences in Fig.3B to clarify the conclusion in line 221-224.

4.      The most significant advantage of uricase-expressing engineered macrophages is reducing the immunogenicity of uricase medications. Therefore, authors should compare the beneficial effects of the new method with rasburicase (mentioned in line 48) beside the comparison with allopurinol – a xanthine oxidase inhibitor.

Reviewer 2 Report

Comments and Suggestions for Authors

In the present manuscript, Feng et al., have shown that the macrophages engineered to release more uricase enzyme are able to mitigate the need for decreasing the uric acid levels in hyperuricemia. The authors have described the compared the impact of these engineered macrophages in reducing uric acid levels in HUA mice model with a known drug allopurinol. They have also considered the impact on other physiological parameters and checked for kidney and liver function test. Overall, the presented work defines the effect of genetic engineering to alleviate uric acid levels in mice.

Although my concerns with this manuscript are as mentioned below.

General concept comments:

1.     Authors should site the concerned recent findings in the field of investigation.

2.     If possible, the authors may mention the ethical approval committee name/number for the local mice experimental protocol approval in the methods section.

3.     Only male mice were used for the study. Hence, authors can comment on the status of reproducibility of such intervention if done in female mice.

4.     In the methodology section, for the “2.5 In Vitro Evaluation of Urate-Lowering Activity” authors can mention the precise volume collected for uric acid analysis.

5.     Authors could have checked or commented on the relative abundance of M0, M1 and M2 specific population of macrophages at the time points they collected the mouse blood.

6.     Authors could have checked or commented on the impact of uricase on blood clotting function in uricase expressing macrophage treated mice.

Reviewer 3 Report

Comments and Suggestions for Authors

Please see the comments attached in pdf file.

Round 2

Reviewer 1 Report

Comments and Suggestions for Authors

All issues have been resolved. The manuscript is now suitable for acceptance.